# Comparison of the Mineral Profile of Two Types of Horse Diet, Silage and Commercial Concentrate, and Their Impacts on Hoof Tensile Strength

**DOI:** 10.3390/ani12223204

**Published:** 2022-11-18

**Authors:** Gabriel Rueda-Carrillo, René Rosiles-Martínez, Luis Corona-Gochi, Anaid Hernández-García, Gabriela López-Navarro, Francisco Trigo-Tavera

**Affiliations:** 1Departamento de Medicina, Cirugía y Zootecnia para Equipos, Facultad de Medicina Veterinaria y Zootecnia, Universidad Nacional Autónoma de México, Av Universidad 3000, Mexico City 04510, Mexico; 2Departamento de Nutrición Animal y Bioquímica, Facultad de Medicina Veterinaria y Zootecnia, Universidad Nacional Autónoma de México, Av Universidad 3000, Mexico City 04510, Mexico; 3Departamento de Genética y Bioestadística, Facultad de Medicina Veterinaria y Zootecnia, Universidad Nacional Autónoma de México, Av Universidad 3000, Mexico City 04510, Mexico; 4Departamento de Patología, Facultad de Medicina Veterinaria y Zootecnia, Universidad Nacional Autónoma de México, Av Universidad 3000, Mexico City 04510, Mexico

**Keywords:** minerals, hoof, tensile, strength, horse, diet

## Abstract

**Simple Summary:**

The hoof is one of the most delicate parts of a horse’s body and its quality depends on a combination of endogenous and exogenous factors. The type of nutrition that a horse receives is a very important factor in determining hoof quality, and a horse’s diet can be supplemented to ensure that it is an adequate source of the energy, proteins, vitamins, and minerals required for healthy growth of all structures comprising the hoof. The aim of this study is to compare two types of horse diets, silage and commercial concentrate, and their impacts on hoof tensile strength.

**Abstract:**

The type of diet that a horse consumes is one of the most important factors determining the mineral profile and tensile strength of its hooves, so the quality and nutritional value of the supplied feed are fundamental to achieving satisfactory hoof quality. The objective of this study was to compare the differences in the mineral concentrations of sodium (Na), potassium (K), magnesium (Mg), zinc (Zn), and Iron (Fe) between two diets, namely silage and a commercial concentrate, using atomic absorption spectrometry and to determine which led to higher increases in the hoof tensile strength of horses consuming these diets. Thirty-two Spanish horses were randomly divided into two groups, where the diet of the horses in group 1 was silage oat hay, and that of the horses in group 2 was a commercial concentrate and oat hay. Both diets were provided for 12 months. The hoof tensile strength was then measured using an Instron universal testing machine. Mg and Fe levels were higher (*p* < 0.05) in the silage than in the commercial concentrate, and hoof Mg, Zn, K, and tensile strength were also influenced by the hormonal stage (*p* < 0.05). The type of diet directly affected the hoof Mg, Zn, K, Na, and tensile strength (*p* < 0.05), which Mg, K and Na were higher in horses fed with concentrate. It was found that for each unit of Zn (μg/g and Na (μg/g)), in the hoof, the tensile strength is affected by −0.10 N/mm^2^ and −0.003 N/mm^2^, respectively for each mineral.

## 1. Introduction

The hoof is one of the most delicate parts of a horse’s body. The stratum medium and outermost stratum externum are responsible for protecting the hoof from external factors such as abrasive, harmful substances, or infectious agents [1]. The quality of the hoof depends on a combination of endogenous and exogenous factors. Endogenous factors include the chemical composition, structure of the stratum corneum, amount and distribution of intracellular fluid, and type of keratin in addition to the amount and arrangement in cells. Exogenous factors include husbandry methods, hoof maintenance with a farrier, diet, climate, and season [2].

The type of nutrition that a horse receives is a very important factor in determining hoof quality, and the diet can be supplemented to ensure that it is an adequate source of the energy, proteins, vitamins, and minerals that ensure healthy growth in all structures comprising the hoof. The quality of hoof structures depends on the availability of necessary minerals such as calcium, zinc, and copper [3,4]. Sources of minerals for horses are typically forages and cereal grains, and their concentration in these sources depends on their concentrations in the soil, forage maturity, plant species, harvest conditions, and conservation methods [5]. According to the National Research Council (NRC), there are 14 essential minerals for equines, seven essential macrominerals (Ca, P, Mg, K, Na, Cl, and S), and six essential microminerals (Zn, Fe, Se, Co, I, and Mn) [6].

Due to environmental conditions and the unavailability/instability of pastures, many horse owners choose to feed their horses preserved forages (hay, haylage, and sometimes silage), which therefore represent all or part of their forage intake [7]. Silage is the result of physical–chemical–biological processes to which a plant is subjected in order to preserve it. Generally, silage is discarded in equine feed due to its low content of vitamins and minerals such as calcium (Ca), phosphorus (P), magnesium (Mg), sulfur (S), chlorine (Cl), sodium (Na), and vitamins of type A, D, and E, but it has been shown that the nutrient content depends on many factors related to the plant and the environment [8]. An additional key factor is the maturity of the forage at the time of harvest [9]. Other reasons for its exclusion from diets are that its acidity is thought to adversely impact health in addition to its hygienic quality being questionable; for this reason, it is recommended that the nutritional quality of silage be characterized by sending a representative sample to the laboratory, in addition to analyzing for the possible presence of pathogenic bacteria, fungi, and mycotoxins [10,11].

It has been shown that supplementing with vitamins, minerals, and amino acids results in improved hoof quality, but there is a need for indicators that allow the implementation of strategies toward optimizing this quality [3]. The objectives of this study were to compare the mineral contents of two diets fed to equines, namely silage and commercial concentrate, in addition to examining the association between mineral content and hoof tensile strength.

## 2. Materials and Methods

### 2.1. Animals

Hoof samples were obtained from 32 Spanish horses that were housed at a facility in Mexico City, with a mean body mass of 429.53 ± 51.23 kg and a mean age of 7.53 ± 2.06 years. The horses were randomly divided into two groups: group 1 (14 horses; 9 stallions and 5 geldings) was fed a diet of silage (3 kg per day) and oat hay (8 kg per day), and group 2 (18 stallions) was fed a diet of commercial concentrate (3 kg per day) (Grano de Oro, Malta Clayton, Oaxaca, Mexico) and oat hay (8 kg per day). Both diets were provided for 12 months. In addition, the hormonal status of the horses (whole or castrated) was considered.

### 2.2. Mineral Components of the Diets

To quantify the mineral elements of the two diets administered to the horses (silage made from corn (50%), alfalfa (25%), and oats (25%) and a commercial feed), samples were taken every 3 months. Samples were processed according to the specifications of the atomic absorption spectrometry equipment (Atomic absorption spectrometer, Perkin Elmer 3100, Waltham, MA, USA). For the protein and fiber crude detection methods, the Official Methods of Analysis, (AOAC 2015; 934.01, 962.09) were used respectively.

### 2.3. Hoof Samples

Equine hoof samples (65 ± 21 g) were obtained from hoof trimmings resulting from shoeing and/or trimming routinely performed on the horses. Separate samples were obtained from the right forelimb, wall area. Prior to trimming, each hoof was cleaned with a wire brush and scrubbed with deionized water. Clippings from each hoof were placed in labeled plastic bags and stored for subsequent analysis [12].

To determine the tensile strength, the samples were cut to a uniform thickness of 0.5 ± 0.1 mm, and subsequently cut to 5 cm × 0.05 cm. Then, using a Dremel tool (Dremel^®^; Mount Prospect, IL, USA), the samples were chiseled in their central area until they were 2 mm thick. The remainder of each hoof clipping was repackaged into a labeled plastic bag and stored at room temperature until processing for mineral analysis [12].

### 2.4. Tensile Strength Hoof Test

The tensile strength of the hoof was measured using an Instron universal testing machine (model 4206, Boston, MA, USA) with a load capacity of 5 kilonewtons (kN) and an acceleration of 2 mm/min; the results are expressed in Newtons (N) as the threshold amount of kg over mm2 necessary to break the sample. This measurement was conducted on a millimeter portion of the hoof cutout (between 10 and 40 mm^2^), which was held by the ends and subjected to tension until breaking [12]). All analyses were performed within 48 h after sampling, since a certain degree of hydration is preserved during this period. The same handling and sampling analysis times were maintained for all samples. The samples contained 99% dry matter.

### 2.5. Mineral Components in the Hoof

The remaining hoof cuttings were used to determine the association between mineral elements (Ca, K, Fe, Mg, Se, Zn, and Na). Firstly, the organic matter was removed followed by pulverizing and digestion with nitric acid at 70 °C for 12 h [12], and analyzed using atomic absorption spectrometry equipment (Perkin Elmer 3100 Atomic Absorption Spectrometer, Waltham, MA, USA). The samples in these studies came from animals that were clipped monthly, so there were insufficient samples from some animals for mineral profiling.

### 2.6. Statistical Analysis

Means and standard errors were obtained for the mineral elements (Ca, Mg, Zn, K, and Na) of both diets (silage and concentrate) and equine hooves in addition to the hoof tensile strength. A comparison was also made between the results obtained by atomic absorption spectrometry and the values mentioned by the NRC through t-testing of means, since the assumptions of normality were met.

### 2.7. Part 1

A linear model with an identity link function was used for a two-factor factorial design, described below:(1)$$Y_{ijk}\;=\;\mu +\alpha_{i}+\beta_{j}+\epsilon_{ijk}$$
where

$Y_{ijk}$ = response variable (Ca, Mg, Zn, K y Na)

μ = population mean

$\alpha_{i}$ = mean of the effects of the diet level i = 1,2,3…

$\beta_{j}$ = average of the effect of hormonal states on the level j = 1,2,3…

$\epsilon_{ijk}$ = experimental error associated with each observation

The assumptions of the model, homogeneity of variance, and normality in the errors, were tested; all models fulfilled the assumptions in the case of the different response variables.

### 2.8. Part 2

To find out which minerals were best associated with tensile strength, a saturated model was used (hoof strength = Ca + Mg + Zn + K + Na) that can be represented as follows:(2)$$y_{i}=\beta_{0}+\beta_{1}X_{i1}+\beta_{2}X_{i2}+.\;.\;.+\beta_{n}X_{in}+\epsilon_{i}$$
where

$y_{i}$ = response variable, hoof strength 

$\beta_{0}$ = is the ordinate to the origin

$\beta_{1}$ … $\beta_{n}$= coefficients

$X_{i1}\ldots \;X_{in}$ = independent variables (Ca, Mg, Zn, K, Na)

$\epsilon_{i}$ = random component with behavior ~N(0,1) 

To choose the optimal model using the AIC, a mixed step-by-step algorithm was used. The final model with AIC = 138.89 was as follows:(3)$$y_{i}=\beta_{0}+\beta_{1}{\left(zn\right)}_{i}+\beta_{2}{\left(na\right)}_{i}+\epsilon_{i}$$
where

$y_{i}$ = response variable (force to tension)

$\beta_{0}\;$ = ordinate to the origin

$\beta_{1}\;y\;\beta_{2}$ = coefficients

${\left(zn\right)}_{i}\;y\;{\left(na\right)}_{i}\;$ = regressors Zn and Na

$\epsilon_{i}$ = random component with behavior ~N(0,1).

The assumptions of linearity, homogeneity of variance, collinearity, and normality in the residuals were tested, and all assumptions were met.

To determine the correlation of the hoof mineral elements and tensile strength with the mineral elements of the feed (concentrate or silage), Pearson correlation testing was performed.

Statistical analyses were performed using the functions lm, glm, step, lmtest, shapiro.test, and Durbin–Watson test of the statistical package R 4.0.2 (Vienna, Austria).

## 3. Results

Table 1 shows the comparison of the means and standard errors of the mineral elements of the two diets administered to the horses against the NRC values, where lower Mg values and higher Fe values were obtained for silage (*p* < 0.05); the mineral levels in commercial concentrate were below those stipulated by the NRC (*p* < 0.05).

Table 2 shows the *p*-values obtained for each variable due to the effect of fixed factors, in the last column the adjusted R^2^ for each variable. It is important to highlight that most of the response variables had an adjusted R^2^ above 80%, which reflects the goodness of fit of the model for the variable to be explained.

Table 3 shows the mineral elements Mg, Zn, K, and tensile strength were influenced by the horses’ hormonal status (*p* < 0.05). The diet factor had an effect on the minerals Mg, Zn, K, and Na and tensile strength (*p* < 0.05)

Table 4 shows the values of the coefficients of the mineral elements that were most strongly associated with hoof tensile strength, where it can be seen that the value obtained for the Zn coefficient was −0.10 N/mm^2^. The same scenario was presented for Na, with a coefficient equal to −0.003 N/mm^2^. The behavior of the two independent variables (Zn and Na) can be seen in Figure 1, where a *p*-value < 0.05 was obtained for both, allowing us to conclude that they were statistically associated in a linear manner.

Figure 2 shows the correlation matrices between the mineral elements of the same administered diets, in addition to the values found in equine hooves. Figure 2A shows the matrix for commercial concentrate, where the following correlations between different mineral elements of the same hoof were observed: Ca was positively correlated with the elements Mg, K, and Na (r = 0.69, 0.48, 0.61, respectively, *p* < 0.05) and, likewise, Mg is positively correlated with K and Na (r = 0.65, 0.62, respectively, *p* < 0.05). Regarding the correlation of particular minerals between hoof and concentrate, the following was observed: hoof Zn was negatively correlated with concentrate Mg (r = −0.05, *p* < 0.05) and hoof K had the same behavior as concentrate Ca (r = −0.89, *p* < 0.05). Figure 2B shows the matrix for silage, where the following correlations between different minerals of the same hoof were observed: Mg was positively correlated with Zn and K (r = 0.82 and 0.55, respectively, *p* < 0.05) and, likewise, K was positively correlated with Na (r = 0.57, *p* < 0.05). Finally, regarding the correlation between silage minerals, Ca was positively correlated with K (r = 0.97, *p* < 0.05).

## 4. Discussion

Through atomic absorption spectrometry analysis, it was found that Mg was lower and Fe was higher in silage, compared to what is presented in the NRC, but it must be considered that forage quality is greatly influenced by the plant environment such as the type of soil, availability of soil nutrients, water, temperature and solar radiation, therefore the end product changes in chemical composition. An improved understanding of the factors that can modify plant development will make it possible to predict the optimal harvest time for forages in order to obtain optimum quality [8,13]. On the other hand, concentrate complies with the requirements of the NRC; low amounts of Cu and no presence of Se were found, Wagner suggest not to supplement them, since he did not find absorption or retention of Cu, Mn or Zn with the use of three different supplements [14].

In contrast to the study carried out by Meyer, where it was found that the absorption of Mg was increased by increasing the amount of fiber [15], it was observed in the present study that the horses fed with concentrate had higher levels of Mg in addition to K and Na, but lower Zn and hoof tensile strength; for this reason, it is important to emphasize that diets based on these foods should always be supplemented with minerals or various forages, depending on their availability in the area. Sadet (2017) found general alterations in pH values and volatile fatty acid concentrations in the cecum, right ventral colon, and feces when adding concentrate to a horse diet compared with a hay diet [16]. In donkeys, it has been suggested that the level of forage in the diet should not be less than 55% to maintain greater use of N and energy in practical feeding, since it was observed that by reducing the forage/concentrate ratio in the diet, protein digestibility and fiber digestibility were significantly increased, which led to markedly decreased N retention as a result of increased urinary N excretion [17]. In contrast, silage-fed horses did not have the same mineral changes as concentrate-fed animals; this situation is similar to that presented by Pagan (1998), where exercised horses that received a forage and grain diet ingested sodium salt blocks to cover their needs, while the animals that consumed only forage did not, thus showing the tendency toward imbalance in animals that consume mixed diets [18]. In any type of diet, the amount of food, hours in which it is offered, and physical activity, as well as the types and forms of food offered should be considered, since they can alter digestion as well as the use of nutrients [19].

We had expected to find associations between the minerals present in the hoof and those of the administered diets, but the mineral elements of the hoof were only correlated with those of the concentrate (Zn with Mg and K with Ca, both negative associations), and in studies such as the one by Domínguez (2017), a correlation was found between the mineral elements of the food and the mineral elements of the animal, but only when the latter were measured in the blood serum [5]. What was mainly observed in the correlation test were associations between the mineral elements of the same hoof, or, in the case of the silage, correlations were obtained between the mineral elements of the same silage; for example, there was a positive association between Ca and K. There are no previous reports where the mineral elements of the diets are associated with the hoof minerals, only studies were found where the mineral elements obtained from digestibility tests through fecal samples, absorption tests through plasma samples and excretion tests through urine and feces samples are related to the mineral elements of the diets [20,21,22,23,24]. Low amounts of Cu were detected in the feed samples analyzed, although, in an investigation carried out in ponies fed with low concentrations of Cu in the diet, it was observed that the retention of this mineral was significantly increased and its absorption at the intestinal level remained stable, in addition, excretion decreased [25].

Geldings presented lower hoof tensile strength than intact males, which may be due to the tendency of intact males to be heavier due to the effects of the hormone testosterone, which is responsible for the structuring of long bones and the disposition of muscle mass [26]. Studies carried out in other animal species have found few differences in mineral elements between hormonal states; for example, the report by Santos of a study on young goats mentions that the amount of P was greater in intact males than in females and castrated males, which suggests that females and castrated males lose and are less efficient at retaining P than castrated males [27]. Ruiz found higher amounts of K and Na in intact male pigs than in castrated males and female pigs [28], and the same was observed in the present work, where castrated males showed higher amounts of Mg and Ca compared to intact males, with no differences in the remaining mineral elements. No additional differences were found in the articles mentioned above. However, when considering the total for all mineral elements in hoof samples, the values in castrated animals are lower than for whole animals, and it is perhaps for this reason that castrated horses are more predisposed to presenting changes in the hoof than their intact counterparts. It is suggested that more relevant studies be carried out to explore this further.

In an exploratory study with 165 horses, a negative correlation was found between Zn and tensile strength [29], and the same was observed in the present study, where a relationship with a coefficient of −0.10 was obtained, and similar results were previously presented in a study on donkey hoof samples, although this was reported as hoof strength [30]. Ley did not find an association between Zn and tensile strength [11], and these findings have to be considered when feeding horses, since zinc is usually supplemented to increase the strength or quality of horse hooves. The same scenario was presented for Na, with a coefficient of −0.003, and for this reason, it is important to pay attention to the diets that are administered to horses, since an imbalance can cause abnormalities in growth and development which predispose the hoof to microfractures [29,31].

## 5. Conclusions

Excessive or low amounts of mineral elements were found in both diets, where the characteristics of the forage are the cause of imbalances in minerals in the resulting silage. Even so, it was observed that there were other factors that caused changes in the mineral elements in hoof samples, such as a horse’s hormonal status (whole or castrated). This factor also influenced the tensile strength of hooves as well as the presence of mineral elements such as Zn and Na. There are still some gaps in current knowledge, regarding the influence of diet’s mineral composition on hoof tensile strength. The present study is to investigate the effects of dietary types on hoof mineral contents and tensile strength, and further research areas are proposed according to some valuable results.

## Figures and Tables

**Figure 1 animals-12-03204-f001:**
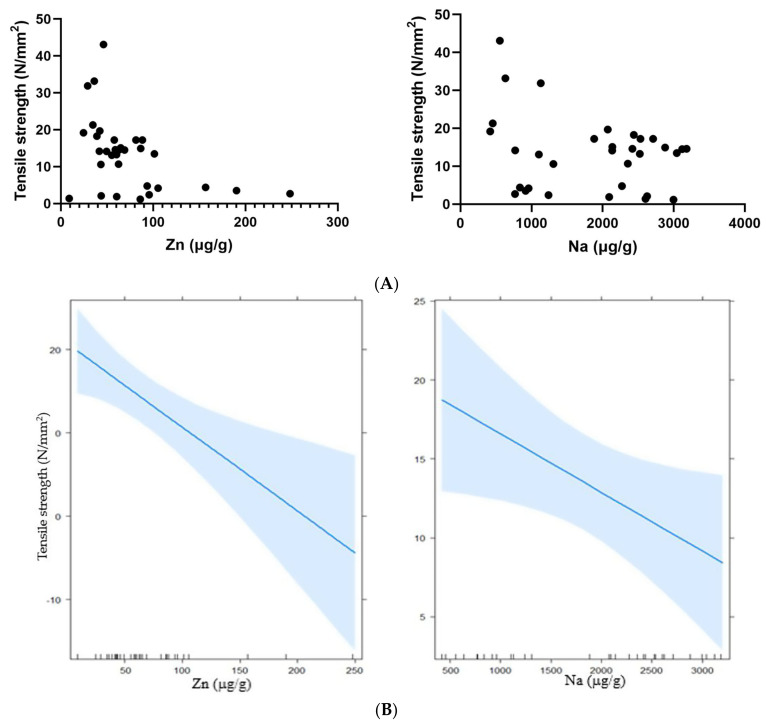
Linear relationship between the mineral elements Na and Zn with the hoof tensile strength. ((**A**) showing the observed values and (**B**) the values estimated by the model).

**Figure 2 animals-12-03204-f002:**
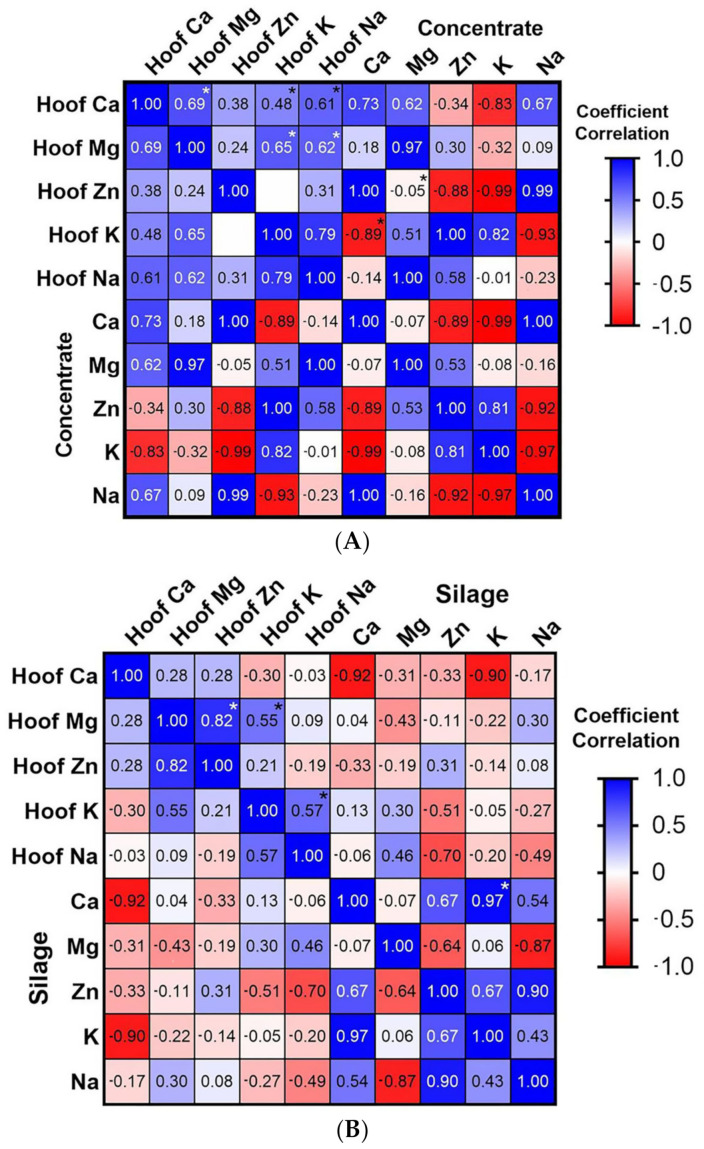
Pearson correlation matrices between mineral elements (μg/mg) of the concentrate (**A**) and silage (**B**) with the horse hoof samples, respectively. * Indicates significant differences at *p* < 0.05.

**Table 1 animals-12-03204-t001:** Comparison of mineral elements (μg/g) in silage, concentrate and oat hay feed samples against NRC values.

Mineral Elements	Silag	Concentrate	Oat Hay	NRC
(μg/g)	(*n* = 4)	(*n* = 3)	(*n* = 3)	
K	1094.25 ± 732.10	1876.00 ± 1262.71	817 ± 111.40	1000
Ca	1156.50 ± 473.37	1916.67 ± 467.20	1233 ± 58.08	800
Na	698.00 ± 215.89	841.33 ± 358.05	428.06 ± 108.03	400
Mg	123.50 ± 3.51 *	216.67 ± 77.78	118 ± 60.4	300
Zn	7.45 ± 6.44	11.63 ± 8.05	17 ± 1.10	16
Fe	103.74 ± 29.07 *	49.80 ± 8.29	89 ± 4.20	16
Mn	16.23 ± 18.72	13.20 ± 2.40	15 ± 12.0	16
Crude protein	9.49%	10%	6.91%	14%
Crude fiber	20.9%	5%	31.44%	16%

* Significant differences at *p* values < 0.05. NRC; National Research Council

**Table 2 animals-12-03204-t002:** Effect of the hormonal status (stallions and geldings) and diet factor, with their respective adjusted R^2^ of the model, for each response variable.

Variable	Sex *p*-Value	Diet *p*-Value	R^2^ Adjusted to the Model
Ca	0.36	0.51	0.95
Mg	0.00	0.00	0.95
Zn	0.00	0.01	0.89
K	0.52	0.00	0.81
Na	0.84	0.00	0.94
Tensile strength	0.00	0.01	0.76

R^2^ = Coefficient of determination; Variables with *p*-values < 0.05 indicate a significant effect of the factor.

**Table 3 animals-12-03204-t003:** Comparison of mineral elements (μg/g) and tensile strength (N/mm^2^) in hoof samples according to hormonal status (stallions and geldings) and type of diet (silage and commercial concentrate).

Variable	Sex		Diet	
	Stallion (*n* = 27)	Gelding (*n* = 5)	Silage (*n* = 14)	Concentrated (*n* = 18)
Ca	618.56 ± 156.79	721.44 ± 63.75	672.51 ± 140.85	605.28 ± 154.63
Mg	170.27 ± 64.93	267.64 ± 35.10 *	162.58 ± 89.64	203.30 ± 46.68 *
Zn	56.51 ± 22.61	159.18 ± 62.97 *	79.61 ± 71.56	67.06 ± 18.49 *
K	1238.37 ± 696.66	1006.00 ± 63.87	869.71 ± 300.75	1460.55 ± 735.09 *
Na	2014.30 ± 884.51	943.74 ± 180.31	978.19 ± 544.53	2522.79 ± 395.82 *
Tensile strength	15.29 ± 9.56	3.44 ± 0.88 *	14.66 ± 13.42	12.50 ± 5.91 *

* Significant differences at *p*-values < 0.05.

**Table 4 animals-12-03204-t004:** Multiple linear regression coefficients for mineral elements and tensile strength.

Independent Variable	Coefficient	EE	CI al 95%	*p*-Value
0	27.60	4.38	18.63, 36.56	0.000
Zn	−0.10	0.03	−0.16, −0.04	0.003
Na	−0.003	0.001	−0.01, 0.00	0.036

## Data Availability

The data presented in this study are available on request from the corresponding author without undue reservation.

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
