# Peer review of "Comparison of the Mineral Profile of Two Types of Horse Diet, Silage and Commercial Concentrate, and Their Impacts on Hoof Tensile Strength"

_animals, 2022, doi:10.3390/ani12223204_

Round 1
Reviewer 1 Report
This manuscript aimed to compare the mineral profile of two types of horse diet, silage and commercial concentrate, and their impacts on hoof tensile strength.
1. Apart from mineral differences, the two diets showed significant differences in other nutrients, and the paper did not give their nutrient levels. The present results cannot be explained only by the mineral differences in the diets.
2. The horses in group one were fed a diet consisting entirely of silage, it's not appropriate.
3. Only the contents of K, Na, Mg, Ca, Zn, Fe and Mn in diets and horse hoof and the hoof tensile strength are considered in this paper, and the results of the present study cannot support the purpose of the trial.
4. The blood mineral concentration is the main parameter reflecting the mineral nutritional status of the body, how are the minerals in the blood?
5. The growth rate of the horse has an impact on the quality of the hoof. Is the growth rate of horses in the two groups the same in this study? Does it affect the quality of the hoof ?
Author Response
This manuscript aimed to compare the mineral profile of two types of horse diet, silage and commercial concentrate, and their impacts on hoof tensile strength.
- Apart from mineral differences, the two diets showed significant differences in other nutrients, and the paper did not give their nutrient levels.
CA: we included on table 1
- The present results cannot be explained only by the mineral differences in the diets.
CA: There are other studies that reported the correlation between the mineral content in the hoof and the tensile strength of it and the importance of minerals in horse feed.
- The horses in group one were fed a diet consisting entirely of silage, it's not appropriate.
CA: The horses were fed with silage and oat hay.
- Only the contents of K, Na, Mg, Ca, Zn, Fe and Mn in diets and horse hoof and the hoof tensile strength are considered in this paper, and the results of the present study cannot support the purpose of the trial.
CA: There are many other studies that correlated the mineral content of the hoof and tensile strength and the important role that they play.
- The blood mineral concentration is the main parameter reflecting the mineral nutritional status of the body, how are the minerals in the blood?
CA: The objective of the study is to know the mineral content that stores in the hoof with two differents types of diets and its correlation with the tensile strength.
- The growth rate of the horse has an impact on the quality of the hoof. Is the growth rate of horses in the two groups the same in this study? Does it affect the quality of the hoof?
CA: Both groups of horses were Spanish horses between 7-9 years old, and 450-500 kg, there were no significant differences between weigh and age, all were adult horses.
Reviewer 2 Report
Originality/Novelty
The question is well defined. The authors contribute new information.
Significance
The results are generally interpreted appropriately and are a contribute to the evaluation of the effect of diet mineral composition on hoof tensile strength. Still, description of associations need to be clarified, especially in the abstract (my specific comment on the astract). The material and methods need to be clarified regarding the specific diets fed to each group.
All conclusions are justified and supported by the results.
Quality of presentation
The article is written in an appropriate way. Data and analysis are presented appropriately.
Scientific soundness
The study seems to be correctly designed and technically sound and the data seem robust enough for the conclusions drawn. Still, the authors need to clarify the two diets used (as in everything that was fed to each group of horses).
Generally, the methods, tools, and software, are described with sufficient details to allow another reseacher to reproduce the results.
Interest to the readers
The conclusions are interesting for the readership of the journal.
Overall Merit
The work provides an advance towards the current knowledge and is relevant for the clinical field.
English level
The English language is appropriate and understandable but needs various corrections. It would be advisable to ask am English native speaker to read the paper.
REVIEW REPORT
Brief summary
The authors aimed at comparing two types of horse diet, silage and commercial concentrate and their impacts on hoof tensile strength by comparing the differences in the mineral concentrations of sodium (Na), potassium (K), magnesium (Mg), zinc (Zn), and Iron (Fe) using atomic absorption spectrometry and by determining which led to higher increases in the hoof tensile strength. The hoof tensile strength 34 was then measured using an Instron universal testing machine. Thirty-two Spanish horses were randomly divided into 32 two groups, where the diet of the horses in group 1 was silage, and that of the horses in group 2 33 was a commercial concentrate. Both diets were provided for 12 months. Mg and Fe levels were higher (P < 35 0.05) in the silage than in the commercial concentrate, and hoof Mg, Zn, K, and tensile strength were 36 also influenced by hormonal stage (P < 0.05). The type of diet directly affected the hoof Mg, Zn, K, 37 Na, and tensile strength (P < 0.05), which were all higher in horses fed with silage; for every increase 38 in unit of Zn (μg/g), the tensile strength decreased at -0.10 N/mm2, and for every unit increase in 39 Na (μg/g), the tensile strength decreased at -0.003 N/mm2.
Broad comments
The authors have addressed a significant topic where there are still some gaps in current knowledge, regarding the influence of diet mineral composition on hoof tensile strength. The present work represents a step forward in this topic, with emphasis on Zn and Na.
The methods used are generaly well described. But some specific comments pointed out underneath should be addressed and the English language should be further revised.
Specific comments
Abstract – “The type of diet directly affected the hoof Mg, Zn, K, Na, and tensile strength (P < 0.05), which were all higher in horses fed with silage; for every increase in unit of Zn (μg/g), the tensile strength decreased at -0.10 N/mm2, and for every unit increase in Na (μg/g), the tensile strength decreased at -0.003 N/mm2.”
This sentence is not clear. The authors say that hoof Mg, Zn, K, Na, and tensile strength, all were higher in horses fed silage, which leads us to think that an increase in hoof Mg, Zn, K and Na is related to an increase in hoof tensile strength. But then the authors say that for every increase in Zn and Na, the hoof tensile strength decreased. Please clarify. I believe that you need to write another small transition sentence between the two sentences.
Line 65 – “Generally, it is discarded in equine feed…” – What does “it” refer to? I believe this sentence needs to be turned around.
Please refrase this sentence: “Generally, it is discarded in equine feed due to its low content of vitamins and 66 minerals such as calcium (Ca), phosphorus (P), magnesium (Mg), sulfur (S), chlorine (Cl), sodium (Na), and vitamins of type A, D, and E, but it has been shown that the nutrient content depends on many factors related to the plant and the environment.”
Line 87 – We only learn now that one of the diets included oat hay. I believe that this should also be mentioned in the abstract.
Line 90 – The authors now tell us that the horses were fed different comercial feeds (concentrate) and no references are provided as to names and manufaturers of these feeds.
I suggest that you state clearly in a table:
- exactly what was fed to each group
- exactly which parts of the diet were analysed in each group
Please also clarify, if you were using different commercial feeds (concentrate), how can you be sure the diferences in mineral componentes between the groups could not also be due to these different commercial feeds.
Line 175 – “horses’ hormonal status” – the authors should make a reference in “Materials and Methods” – “Animals” of what the horses’ hormonal status is considered as…gelding or stallion.
Line 223 – This sentence needs to be corrected.
Suggestion: “Through atomic absorption spectrometry analysis, it was found that Mg was lower and Fe was higher in silage, compared to what is presented in the NRC, but it must be considered that forage quality is greatly influenced by the plant environment such as type of soil, availability of soil nutrients, water, temperature and solar radiation, therefore the end product changes in chemical composition.”
Line 232 - Please substitute “here” by “in the present study”
Line 276 – “… the values were in castrated animals are lower than for in whole animals, …”
Line 277 – Please substitute “alterations” by “changes”
Line 274-278 – This sentence is too long. Please divide i tinto smaller sentences:
“However, when considering the total for all mineral elements in hoof samples, the values were in castrated animals are lower than for whole animals, and it is perhaps for this reason that castrated horses are more predisposed to presenting alterations in the hoof than their intact counterparts, and it is suggested that more relevant studies be carried out to explore this further.”
Line 292 - Please substitute “alterations” by “changes”
Conclusion – I suggest that the authors re-write the conclusion in a way that the ideas are more systematically organized.
RECOMMENDATION
Accept after minor revision and English language revision
Author Response
The authors have addressed a significant topic where there are still some gaps in current knowledge, regarding the influence of diet mineral composition on hoof tensile strength. The present work represents a step forward in this topic, with emphasis on Zn and Na.
The methods used are generaly well described. But some specific comments pointed out underneath should be addressed and the English language should be further revised.
The paper was submitted for English editing to the MDPI [Manuscript ID: English-49799] MDPI English editing
Specific comments
Abstract – “The type of diet directly affected the hoof Mg, Zn, K, Na, and tensile strength (P < 0.05), which were all higher in horses fed with silage; for every increase in unit of Zn (μg/g), the tensile strength decreased at -0.10 N/mm2, and for every unit increase in Na (μg/g), the tensile strength decreased at -0.003 N/mm2.”
This sentence is not clear. The authors say that hoof Mg, Zn, K, Na, and tensile strength, all were higher in horses fed silage, which leads us to think that an increase in hoof Mg, Zn, K and Na is related to an increase in hoof tensile strength. But then the authors say that for every increase in Zn and Na, the hoof tensile strength decreased. Please clarify. I believe that you need to write another small transition sentence between the two sentences.
CA: we modify the sentence.
Line 65 – “Generally, it is discarded in equine feed…” – What does “it” refer to? I believe this sentence needs to be turned around.
Please refrase this sentence: “Generally, it is discarded in equine feed due to its low content of vitamins and 66 minerals such as calcium (Ca), phosphorus (P), magnesium (Mg), sulfur (S), chlorine (Cl), sodium (Na), and vitamins of type A, D, and E, but it has been shown that the nutrient content depends on many factors related to the plant and the environment.”
CA: we did it
Line 87 – We only learn now that one of the diets included oat hay. I believe that this should also be mentioned in the abstract.
CA: we mentioned also in the abstract, thanks.
Line 90 – The authors now tell us that the horses were fed different comercial feeds (concentrate) and no references are provided as to names and manufaturers of these feeds.
CA: the manufacturer of the feed was included in materials and methods.
I suggest that you state clearly in a table:
- exactly what was fed to each group
- exactly which parts of the diet were analysed in each group
CA: this information is on table 1
Please also clarify, if you were using different commercial feeds (concentrate), how can you be sure the diferences in mineral componentes between the groups could not also be due to these different commercial feeds.
CA: it was only one commercial feed, the silage and the concentrate differences are described in table 1.
Line 175 – “horses’ hormonal status” – the authors should make a reference in “Materials and Methods” – “Animals” of what the horses’ hormonal status is considered as…gelding or stallion.
CA: we included in materials and methods
Line 223 – This sentence needs to be corrected.
Suggestion: “Through atomic absorption spectrometry analysis, it was found that Mg was lower and Fe was higher in silage, compared to what is presented in the NRC, but it must be considered that forage quality is greatly influenced by the plant environment such as type of soil, availability of soil nutrients, water, temperature and solar radiation, therefore the end product changes in chemical composition.”
CA: we changed, thanks for the suggestion.
Line 232 - Please substitute “here” by “in the present study”
CA: we did it
Line 276 – “… the values were in castrated animals are lower than for in whole animals, …”
CA: we did it
Line 277 – Please substitute “alterations” by “changes”
CA: we did it
Line 274-278 – This sentence is too long. Please divide i tinto smaller sentences:
“However, when considering the total for all mineral elements in hoof samples, the values were in castrated animals are lower than for whole animals, and it is perhaps for this reason that castrated horses are more predisposed to presenting alterations in the hoof than their intact counterparts, and it is suggested that more relevant studies be carried out to explore this further.”
CA: we did it
Line 292 - Please substitute “alterations” by “changes”
CA: we did it
Conclusion – I suggest that the authors re-write the conclusion in a way that the ideas are more systematically organized.
CA: we did it
Reviewer 3 Report
Dear Authors, please include the marked words!

Author Response
Dear Reviewer,
We include the marked words in yellow.
Thanks
Round 2
Reviewer 1 Report
Reviewer's comments:
This manuscript explained and modified the questions raised by the reviewers, but there are still the following problems to be considered:
Major concerns:
1. Authors indicated that: The horses were fed a diet of silage and oat hay, and a diet of commercial concentrate and oat hay. Therefore, the mineral content of the diet provided in table 1 should include the contribution of oat grass when compared to the NRC. What is the proportion of silage and oat in the first diet? What is the proportion of concentrate feed and oat grass in the second diet? This information should be supplemented on the manuscript. Although the manuscript mentioned related information (To quantify the mineral elements of the two diets administered to the horses (silage made from corn (50%), alfalfa (25%), and oats (25%) and a commercial feed), samples were taken every 3 months. ) , the description of the article is not clear. Oat hay is not included in the diet composition.
2. Does the supplemental fiber in Table 1 refer to NDF or crude fiber or…? Protein and fiber detection methods should be supplemented in materials and methods.
3. There are ……. and hoof minerals, “since digestibility studies on mineral elements are mainly carried out on urine, feces, and blood samples” , the content of this sentence is not well expressed. Is the digestibility research carried out on urine and blood ?
4. Further research should focus on the relation on type of diet, horse individual characteristics, hoof mineral contents and tensile strength. What does the writer mean? The present study is to investigate the effects of dietary types on hoof mineral contents and tensile strength, and further research areas are proposed according to some valuable results.
Author Response
Dear Reviewer,
You can find the answers within the text.
- Authors indicated that: The horses were fed a diet of silage and oat hay, and a diet of commercial concentrate and oat hay. Therefore, the mineral content of the diet provided in table 1 should include the contribution of oat grass when compared to the NRC. What is the proportion of silage and oat in the first diet? What is the proportion of concentrate feed and oat grass in the second diet? This information should be supplemented on the manuscript. Although the manuscript mentioned related information (To quantify the mineral elements of the two diets administered to the horses (silage made from corn (50%), alfalfa (25%), and oats (25%) and a commercial feed), samples were taken every 3 months.) , the description of the article is not clear. Oat hay is not included in the diet composition.
CA: we clarify that and included the oat hay information ( table 1), and the diets proportions for each group.
- Does the supplemental fiber in Table 1 refer to NDF or crude fiber or…? Protein and fiber detection methods should be supplemented in materials and methods.
CA: Fiber in Table 1, refers to crude fiber. Protein and fiber detection methods are supplemented in materials and methods. AOAC, 2015 ( 934.01, 962.09).
- There are ……. and hoof minerals, “since digestibility studies on mineral elements are mainly carried out on urine, feces, and blood samples” , the content of this sentence is not well expressed. Is the digestibility research carried out on urine and blood ?
CA: we corrected, “There are no previous reports where the mineral elements of the diets are associated with the hoof minerals, only studies were found where the mineral elements obtained from digestibility tests through fecal samples, absorption tests through plasma samples and excretion tests through urine and feces samples are related to the mineral elements of the diets [20, 21, 22, 23, 24].”
- Further research should focus on the relation on type of diet, horse individual characteristics, hoof mineral contents and tensile strength. What does the writer mean? The present study is to investigate the effects of dietary types on hoof mineral contents and tensile strength, and further research areas are proposed according to some valuable results.
CA: we refer to type of diet, horse individual characteristics (gelding or stallion), hoof mineral contents and tensile strength.
We included this sentence: “The present study is to investigate the effects of dietary types on hoof mineral contents and tensile strength, and further research areas are proposed according to some valuable results”.
Thanks!